# Management of Chronic Hepatitis B in HIV-Coinfected Patients

**DOI:** 10.3390/v14092022

**Published:** 2022-09-13

**Authors:** Massimo Fasano, Maria Cristina Poliseno, Josè Ramon Fiore, Sergio Lo Caputo, Antonella D’Arminio Monforte, Teresa Antonia Santantonio

**Affiliations:** 1Infectious Diseases Unit, Ospedale della Murgia “F. Perinei”, 70022 Altamura, Italy; 2Clinic of Infectious Diseases, Department of Clinical and Surgical Sciences, University of Foggia, 71122 Foggia, Italy; 3Clinic of Infectious and Tropical Diseases, ASST Santi Paolo e Carlo, University of Milan, 20142 Milan, Italy

**Keywords:** hepatitis B virus, hepatitis B, chronic hepatitis B, hepatocellular carcinoma, coinfection HIV-HBV, treatment, human immunodeficiency virus

## Abstract

Hepatitis B virus infection occurs in approximately 7% of people living with HIV (PLWH), with substantial regional variation and higher prevalence among intravenous drug users. Early studies on the natural history of HIV/HBV coinfection demonstrated that in coinfected patients, chronic hepatitis B (CHB) has a more rapid progression than in HBV-monoinfected patients, leading to end-stage liver disease complications, including hepatocellular carcinoma. Therefore, the adequate management of CHB is considered a priority in HIV-coinfected patients. Several guidelines have highlighted this issue and have provided recommendations for preventing and treating HBV infection. This article discusses the management of liver disease in patients with HIV/HBV coinfection and summarizes the current and future therapeutic options for treating chronic hepatitis B in this setting.

## 1. Introduction

Human immunodeficiency virus (HIV) and hepatitis B virus (HBV) are major public health problems worldwide. According to the WHO, approximately 240 million people are chronically infected with HBV [1], and 37 million people were estimated to be living with HIV by the end of 2020 [2]. HBV shares with HIV the same transmission routes (sexual, parenteral, and vertical transmission), therefore HIV/HBV coinfection is relatively common among people living with HIV (PLWH) [3,4]. Several studies on the natural history of HIV/HBV coinfection have shown more rapid and frequent progression to cirrhosis, higher rates of hepatocellular carcinoma (HCC) development, and liver-related death in HIV/HBV coinfected patients than in those with HBV alone [5,6,7,8]. Despite advances in HBV prevention and care in the past few decades, HIV/HBV coinfection remains a significant cause of non-AIDS-related death and liver-related complications [9]. Therefore, it is crucial to test and treat all PLWH for chronic HBV infection.

This report reviews the management of liver disease in patients with HIV/HBV coinfection and summarizes the current and future therapeutic options for treating chronic hepatitis B.

## 2. Prevalence of HIV/HBV Coinfection

Globally, the prevalence of HIV/HBV coinfection among PLWH is approximately 7.4% [2,4], with substantial regional variation and with higher prevalence (11.8%) among people who inject drugs [10]. In Western Europe and North America, the prevalence of HIV/HBV-coinfection is reported to be 5%, whereas in developing countries it ranges from 2% to 30.6% [2].

In the Italian Cohort of Antiretroviral Naïve Patients (ICONA cohort), a wide cohort of more than 19,000 HIV-positive individuals enrolled between 1997 and 2021 by 55 centers operating throughout Italy, the proportion of HBsAg-positive subjects was 6.6%, with a decreasing trend from 7.1% in 1997 to 4.5% in 2021 [11]. Coinfection prevalence was higher among men (6.6%) than among women (4.6%) and varied according to the mode of HIV transmission. In particular, the prevalence was 8.8% (out of 2685) among injecting drug users, 5.8% (out of 6716) among heterosexual contacts, 5.3% (out of 6635) among homo/bisexual contacts, and 6.4% (out of 1448) among those with other/unknown modes of HIV transmission [11].

## 3. Management of HIV/HBV Coinfected Patients

European and international guidelines recommend testing all PLWH for HBV and vice versa because of the same transmission routes [12,13,14,15]. Initial HBV testing includes HBsAg, anti-HBs, and anti-HBc [13].

Three serological profiles can be identified:HBsAg-negative, anti-HBs-negative, and anti-HBc-negative: PLWH without HBV infection.HBsAg-negative, anti-HBs-negative/positive, and anti-HBc-positive: PLWH with a past HBV infection.HBsAg-positive, anti-HBs-negative, anti-HBc-positive: PLWH with ongoing HBV infection.

These three categories of HIV-positive subjects have different prognoses and treatment indications and require different monitoring and prevention measures (Figure 1).

## 4. HBV Seronegative Patients

All PLWH who are HBV seronegative (HBsAg-negative, anti-HBs-negative, and anti-HBc-negative) and then susceptible to HBV infection should receive hepatitis B vaccination. In adults with HIV infection, HBV vaccination response to the recombinant vaccines is significantly lower than that in healthy adults who are HIV negative [16]. Higher doses (40 μg) or reinforced vaccines (four doses) or adjuvanted vaccines have been suggested to achieve a protective anti-HBs response in these immunocompromised patients [13,17,18]. Undetectable or minimum HIV RNA viral load and higher CD4 cell count at baseline are the factors most frequently associated with a successful response to HBV vaccination in both randomized controlled trials and observational studies [19].

## 5. Subjects with a Past HBV Infection

“Occult” HBV Infection (OBI) is characterized by the persistence of the viral minichromosome defined as covalently closed circular DNA (cccDNA) in the nucleus of hepatocytes. In subjects with a past HBV infection, the replication activity of cccDNA and viral protein expression is strongly suppressed by the host’s immune response [20]. OBI is recognized as one of the possible phases in the natural history of chronic HBV infection [14]. People with OBI can experience the reactivation of HBV replication if the immunological control declines. In HIV-positive patients, OBI reactivation is not frequently reported [21] and has become negligible in those receiving appropriate ART therapy. However, HBV reactivation can still occur in coinfected patients when antiretroviral regimens containing drugs active against HBV are withdrawn.

In fact, Salpini and colleagues showed that HBV virological reactivation is a frequent event in HBsAg-negative/anti-HBc-positive PLWH switching to TDF/TAF sparing therapy [22]. The study included 101 HBsAg-negative/anti-HBc-positive patients with HIV infection from the ICONA Cohort, virologically suppressed for HIV > 12 months. Patients were tested for HBV DNA and HBV RNA within 12 months after the therapeutic switch. After TDF/TAF withdrawal, HBV virological reactivation occurred in a relevant proportion (40%) of patients. A longer duration of TDF/TAF withdrawal correlated with a progressive increase in serum HBV-DNA, supporting the progressive enhancement of HBV replicative activity. The risk of virological HBV reactivation (HBV-R) was associated with the status of HBV reservoir at T0 (still under TDF/TAF based cART), with a higher risk in patients with cryptic HBV-DNA at T0 (serum HBV-DNA > 1 IU/mL). Low levels of anti-HBs (<100 mIU/mL) were the only factor correlated with cryptic HBV-DNA by multivariate analysis. Moreover, HBV reactivation was modulated by the extent of HIV-mediated immunocompromission. In fact, lower nadir CD4+ T cell counts (<100 cells/mm^3^) correlated with an increased risk of HBV-R, as confirmed by multivariable analysis, suggesting that a weakened immune response is a driver for re-uptake/enhancement of HBV replicative activity [22].

These findings highlight the importance of a proper screening and continuous monitoring of HBsAg-negative/anti-HBc-positive patients with HIV infection (particularly if considered for a therapeutic switch not including TDF/TAF), and that ultrasensitive assays for serum HBV-DNA, with novel HBV biomarkers (HBV-RNA, quantitative anti-HBc), can optimize the management of HBsAg-negative/anti-HBc-positive patients with HIV infection.

Additionally, HBV reactivation with consequent development of hepatitis B may occur in patients undergoing immune-suppressive therapies [23]. Therefore, anti-HBc-positive PLWH, untreated with HBV-active antiretroviral therapy (ART), but receiving low-risk immunosuppressive therapy, should be monitored for HBV-DNA and HBsAg for early detection of HBV reactivation while patients treated with potent immunosuppressive therapy (B-cell-depleting agents for lymphoma/leukemia or stem-cell or solid organ transplantation) should receive TDF/TAF therapy to prevent HBV reactivation.

## 6. HBsAg Positive PLWH

In PLWH with ongoing HBV infection, it is essential to establish the phase of the infection and to assess the activity and severity of the liver disease. For this purpose, the initial evaluation of HBsAg-positive patients should include HBeAg, anti-HBe, and serum HBV DNA levels, a complete medical history, a physical examination, and biochemical parameters (complete blood count, ALT, AST, GGT, ALP, hepatic synthetic function (e.g., coagulation, albumin, cholinesterase). Liver fibrosis should be determined by liver biopsy or by non-invasive methods, which include liver stiffness measurements and serum biomarkers of liver fibrosis. Additionally, an abdominal hepatic ultrasound should be part of the initial evaluation in all patients. During follow-up, coinfected patients with liver cirrhosis should undergo serial liver ultrasound examinations and alpha-fetoprotein serology every 6 months for HCC screening. Coinfection with hepatitis D virus (HDV) and/or hepatitis C virus (HCV) should be excluded, as well as alcohol, metabolic liver disease with steatosis or steatohepatitis, and other causes of chronic liver disease [12,13,14,15].

Several studies on the natural history of HIV/HBV coinfection have demonstrated a worse prognosis compared to HBV-monoinfected patients: higher HBV replication levels, more rapid and frequent progression to cirrhosis, hepatic decompensation, HCC development, and liver-related mortality [5,6,7,8].

To prevent liver disease progression and its complications, thus improving survival and quality of life, national and international guidelines recommend treating all HIV-positive subjects with chronic HBV infection [12,13,14,15].

## 7. Current Treatment Strategies

Two classes of drugs are currently approved for treating HBV monoinfected patients: pegylated interferon-alpha (Peg-IFN) and nucleoside/nucleotide analogs (NAs). NAs are oral direct antiviral agents that inhibit the HBV polymerase/reverse transcriptase (RT), an enzyme with a crucial role in the HBV life cycle, as it reversely transcribes the pregenomic RNA into DNA. HBV polymerase/reverse transcriptase has high homology to the reverse transcriptase of retroviruses, such as HIV [24], and most of the NAs were first investigated as a form of antiretroviral therapy.

Entecavir (ETV), tenofovir disoproxil fumarate (TDF), and tenofovir alafenamide (TAF) are the third-generation NAs recommended as first-line monotherapy by all international guidelines for treating CHB due to their high potency and barrier to resistance development [14,15]. They block the production of new virions and progressively reduce serum HBV DNA to undetectable levels, but have little or no effect on the cccDNA in the hepatocyte nuclei. The persistence of the intrahepatic cccDNA determines the reactivation of HBV replication after interrupting NA treatment, thereby justifying the need for long-term therapy for sustained viral replication control.

Emtricitabine (FTC), lamivudine (3TC), TDF, and TAF are components of ARV regimens with activity against HBV. TDF and TAF have a high genetic barrier for the development of resistance mutations. Entecavir is an HBV nucleoside analog that also has weak HIV activity. Guidelines recommend that all people with HIV and hepatitis B co-infection should use combination antiretroviral therapy containing TDF or TAF plus either lamivudine or emtricitabine [12,13,14,15]. ART should be started as soon as possible, regardless of CD4 cell count and HBV DNA level [12,13].

TAF is preferred over TDF in PLWH with established renal and bone impairment. Entecavir may be prescribed in PLWH with no prior exposure to lamivudine and with fully active ART if TAF is strictly contraindicated. Peg-IFN monotherapy is not considered anymore, as ART should be given to all PLWH independently from CD4 count [12,13], (Table 1).

## 8. Monitoring of the Response to Therapy and Adverse Events

All treated patients should undergo periodic monitoring for efficacy and safety. Guidelines recommend monitoring liver blood tests every 3 months during the first year and every 6–12 months after that. Serum HBV-DNA should be determined every 3–6 months during the first year and every 12 months after that. HBsAg should be checked at 12-month intervals at least until the loss of HBsAg. Drug toxicity (renal, bone density, liver) should be closely monitored, and the dose of antiretroviral drugs should be adjusted in the presence of impaired renal function. Due to the high risk of HBV reactivation, stopping TDF- or TAF-containing ART should be avoided in patients with HIV/HBV coinfection, and eventually considered after confirmed HBsAg clearance [12,13].

## 9. Efficacy and Safety Data from Real-Life Clinical Practice

In HBV-monoinfected patients, long-term ETV or TDF monotherapy prevents the progression of liver disease and is associated with the regression of liver fibrosis [14]. Moreover, in cirrhotic patients, the virological response is associated with a reduction in decompensation and an improvement in portal hypertension, thus reducing the need for liver transplantation [25,26]. However, HCC may still develop and remains the major complication for CHB patients treated with NAs [27,28], although a recent study reported a decreasing incidence after the first 5 years of ETV or TDF therapy in CHB [29].

The clinical benefits of HBV antiviral therapy in HIV/HBV coinfected patients remain unclear [9].

In a French prospective cohort study, liver fibrosis evolution was evaluated in 167 HIV/HBV coinfected patients on TDF-containing ART followed for a median period of 60 months. Liver fibrosis levels remained stable in most coinfected patients, and even more fibrosis progression was still observed in 17% of virologically suppressed patients [30].

In a large retrospective cohort study among 3573 HIV/HBVcoinfected patients enrolled in ten American and Canadian cohorts of the North American AIDS Cohort Collaboration on Research and Design, 111 (3%) developed liver complications (90 with ESLD, 11 with HCC, and 10 with both) [31]. Higher baseline fibrosis-4 (FIB-4) score, lower CD4 cell count, and diabetes mellitus were associated with a greater risk of liver complications during follow-up. The risk of liver complications was reduced in subjects who received HBV-active ART, but the results were not statistically significant. HIV suppression for ≥6 months was associated with a lower risk of liver complications [31].

Another large study evaluated liver disease progression and all causes of hospital mortality among 48,189 patients discharged with CHB from French hospitals between 2008 and 2013 [32]. HIV coinfection was reported in 5757 (12%) patients. Liver disease progression was documented in 7479 (15.5%) patients, including end-stage liver disease or HCC. Liver transplantation and death were recorded in 433 (8.2%) and 5299 (11%) patients. Liver-related risk factors other than CHB, such as hepatitis D virus coinfection, hepatitis C virus coinfection, alcohol abuse, diabetes mellitus, and other causes of liver damage, were associated with liver disease progression and increased the risk of all-cause mortality. Despite the higher prevalence of liver-related risk factors, HIV coinfected patients without AIDS had a better outcome. The authors suggested that treatment of CHB and the better control of chronic hepatitis B-associated risk factors is more efficient in coinfected patients with HIV [32].

The long-term outcome of anti-HBV strategies in HIV-positive patients has been evaluated in 634 HIV/HBV coinfected patients from the ICONA cohort, according to the use of anti-HBV drugs (ART with no activity against HBV = 357; lamivudine/emtricitabine = 86; tenofovir plus lamivudine/emtricitabine = 140; tenofovir or entecavir = 41). Both mono- and dual therapy provide benefits compared to no ant-HBV therapy [33].

Additional studies are warranted to verify the long-term benefits of HBV-active ART on liver disease progression and liver-related mortality in HIV/HBV coinfected patients.

## 10. Future Treatment Options

Current therapies (ETV, TDF, or TAF) are effective, safe, inexpensive, and easy to manage, but rarely achieve a functional cure, defined as a sustained off-treatment loss of HBsAg with or without the acquisition of anti-HBs, and undetectable HBV DNA after a finite course of therapy.

In monoinfected patients, continuous treatment with ETV or TDF for up to 10 years results in overall HBsAg seroclearance rates of 0–5%, with higher rates in HBeAg-positive patients [34].

The rate of functional cure in HIV/HBV coinfected patients during ART is not well defined. In a large HIV/HBV cohort study conducted in Zambia, 29/284 (10.2%) HIV/HBV coinfected patients achieved HBsAg loss (functional cure) within 2 years of tenofovir-based ART. In multivariable analysis, CD4 < 350 cells/mm^3^, female sex, and lower baseline HBV DNA levels were associated with greater odds of a functional cure [35].

A recent Dutch multicenter cohort study investigated the kinetics of HBsAg in 104 HIV/HBV coinfected patients treated with TDF-based ART for a median duration of 57 months. A decline of 2.2 log IU/mL in HBsAg levels was observed among HBeAg–positive patients, whereas HBeAg-negative patients only achieved a decline of 0.6 log IU/mL during 6 years of TDF therapy. HBsAg decline was correlated with an increased CD4 cell count, underlining the importance of immune restoration in HBV clearance [36].

Based on the results of these two studies, the HBsAg clearance during treatment seems more easily achievable in HIV/HBV coinfected patients than in HBV monoinfected patients.

HBsAg loss indicates profound suppression of HBV replication and viral protein expression and allows safe discontinuation of antiviral therapy in HBV mono-infection. Moreover, HBsAg loss improves survival and lowers HCC incidence in patients who are on NA therapy [37]. Therefore, there is great interest in developing finite treatment strategies that will achieve durable off-treatment clearance of HBsAg and HBV DNA. This will probably require therapeutic strategies that target the virus as well as the host immune response [38]. The new antiviral agents that have reached clinical development include entry inhibitors, capsid assembly modulators, inhibitors of subviral particle release, and RNA interference molecules. Immunomodulators, namely innate immunomodulators (Toll-like receptor agonists), therapeutic vaccines, checkpoint inhibitors, and monoclonal antibodies, are also progressing toward clinical development [38]. The efficacy and safety of new combinations of antivirals with or without immunomodulators are currently being investigated in clinical trials in HBV monoinfected patients.

The clinical implications of these new therapeutic options in HIV/HBV coinfected individuals, in whom ART must be continued for HIV, are currently unknown. Further studies are needed to clarify the role of curative HBV therapies to maximize effectiveness in coinfected HIV/HBV patients.

## 11. Conclusions

Despite the availability of safe and effective vaccines and HBV active ART, HIV/HBV coinfection remains a significant factor in liver-related complications and non-AIDS-related death. Future efforts should aim at identifying patients at the greatest risk of disease progression and at defining new hepatitis B treatment strategies capable of acquiring a functional cure in most treated patients.

## Figures and Tables

**Figure 1 viruses-14-02022-f001:**
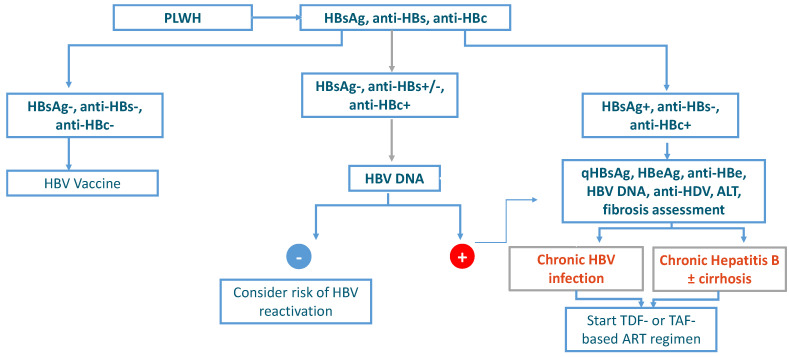
Algorithm for the management of HBV infection in PLWH.

**Table 1 viruses-14-02022-t001:** Treatment of HBV in patients with and without HIV infection.

	Regimen	Drugs	Duration	Route of Administration
**HBV monoinfection**	Monotherapy	ETV, TDF, TAF	Long-term until HBsAg loss	Oral
	Monotherapy	Peg-IFN	48 weeks	Subcutaneous injection
**HIV/HBV coinfection**	Combination regimen	ART containing TDF/TAF plus emtricitabine or TDF/TAF plus lamivudine ART plus ETV	Indefinitely	Oral

## Data Availability

The data used for this review are published articles that are open public on the web.

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
