# Peer review of "Management of Chronic Hepatitis B in HIV-Coinfected Patients"

_viruses, 2022, doi:10.3390/v14092022_

Round 1

Reviewer 1 Report

This is a short review article on the management of chronic hepatitis B in HIV patients. Authors divided coinfected people into 3 groups (those without prior exposure, those with previous infection, and those with active HBV) based on HBV serologies and elaborated treatment plans accordingly for each group.

Other comments:

Introduction has multiple sentence typos (indentation even if not start of sentences) that needs fixing.

Introduction 1st paragraph – spell out PLWH when first used in the main manuscript.

How does sharing same transmission route increase frequency of HBV reactivation and occult HBV infection?

In Section 5 (Subjects with a past HBV infection) – what is the rate of HBV reactivation in PLWH for OBI cases? Is the rate in ICONA study of 40% HBV-R independent of patient’s anti-HBs antibody titer status?

Section 6, 1st paragraph – Liver fibrosis should be determined by liver biopsy or invasive methods? Or did you mean “non-invasive methods”?

I agree that abdominal US should be checked but authors need to add when and how often and for what purpose.

Section 7 – should discuss all ARTs with activity against HBV (omitted were 3TC and FTC) and discuss issues and nuances.

Comment on Section 10 - Even if a HIV-HBV coinfected patient achieve functional cure, he/she will still need to take ARTs for his HIV and be kept on simple combo regimens such as BIC/TAF/FTC.

Lastly, a figure summarizing the management plans would be helpful to the readers.

Reviewer 2 Report

I enclose your PDF in which I have included some comments for changes to be considered.

Reviewer 3 Report

Review of Viruses: 1857225: Management of Chronic Hepatitis B infection in HIV-co-infected Patients

The investigators, Peliseno et al.  have reviewed the current research on clinical management of patients with chronic HBV infection and or co-infected with HIV and recommend appropriate antiretroviral therapies for this population. 

It is a well-written paper discussing the currently available therapies for HIV and HIV/HBV coinfected patients and suggesting appropriate antiviral treatment protocols for patients that are infected with not only HBV, but also for those who are co-infected with HCV, that is known to exacerbate hepatic fibrosis from HBV, which is also true for HCV. 

Although the investigators have discussed the complications of co-infections of HBV and HIV among people who use illicit drugs (IDUs), it would have been appropriate to also discuss the problems of clinical management of IDUs co-infected with HIV, HCV and HBV, and then possibly discuss/recommend treatment modalities for IDUs co-infected with mono-and multiple infections.

 In addition, it would be better of the investigators would consider tabulating treatment protocols for those patients who are mono-infected and co-infected with HIV, HBV etc. 

A minor suggestion: the investigators may want to write out the details of antiretroviral medicine like TDF (tenofovir disoproxil fumarate), TAF (tenofovir alafenamide) and ETC (entecavir), when first mentioned in the paper; and similarly, describe ICONA cohort (Italian Cohort of Patients Naive from Antiretrovirals), when first mentioned in the paper.

Recommendations: Minor revision. 

Round 2

Reviewer 1 Report

Section 7, 3rd paragraph, 1st sentence: Revise to "Emtricitabine (FTC), lamivudine (3TC), TDF, and TAF are components of ARV regimens with activity against HBV." as FTC is not by itself approved for HBV.

Section 7, 3rd paragraph, 3rd sentence with typo - "Entecavi" correct to Entecavir.

Author Response

  • Section 7, 3rd paragraph. The 1st sentence has been changed according reviewer's suggetion.
  • "Entecavi" has been corrected

Reviewer 2 Report

All suggestions have been done. There is one little mistake in section 7 (line 18) Entecavi must be changed by entecavir.

Author Response

"Entecavi" has been corrected